# Effect of Green Human Resource Management on Green Psychological Climate and Environmental Green Behavior of Hotel Employees: The Moderator Roles of Environmental Sensitivity and Altruism

Fatih Uslu [1], Ali Keles [2,*], Arif Aytekin [3], Ozgur Yayla [4], Huseyin Keles [5], Gozde Seval Ergun [6] and Abdullah Tarinc [7]

1    Department of Educational Sciences, Faculty of Education, Akdeniz University, Antalya 07600, Türkiye
2    Linda Hotel, Antalya 07600, Turkey
3    Department of Social Work, Manavgat Social Sciences and Humanities Faculty, Akdeniz University, Antalya 07600, Türkiye
4    Department of Recreation Management, Manavgat Tourism Faculty, Akdeniz University, Antalya 07600, Türkiye
5    Department of Tourism Guidance, Manavgat Tourism Faculty, Akdeniz University, Antalya 07600, Türkiye
6    Department of Tourism Management, Manavgat Tourism Faculty, Akdeniz University, Antalya 07600, Türkiye
7    Department of Gastronomy and Culinary Arts, Manavgat Tourism Faculty, Akdeniz University, Antalya 07600, Türkiye
*    Correspondence: alikelesh.mail@gmail.com; Tel.: +90-5334877570

**Abstract:** This article reports the findings of how green human resource management (GHRM) practices can influence the perceptions of hotel employees regarding their organizations' commitments to green psychological climate (GPC) and their environmentally responsible behavior. GHRM practices refer to the policies and procedures that support environmental sustainability and reduce the negative effects of business activities on the environment. The data used in the research were collected from 425 employees working in 11 5-star hotels in the Antalya/Manavgat districts. For data analysis, data scan analysis was used and the results were then analyzed through the AMOS software to test the structural model. The study has suggested that GHRM practices can positively affect the perceptions of employees concerning their commitment to GPC, which in turn can lead to more environmentally green behaviors. The study also examines the roles of environmental sensitivity and altruism in the relationship between GHRM practices and environmental behavior. The research has shown that individuals with higher levels of environmental sensitivity and altruism are more likely to exhibit environmentally responsible behavior. This means that the employees who are sensitive to environmental issues and have an altruistic personality are more liable to respond positively to GHRM practices of their businesses and are more likely to be environmentally responsible. The research has also emphasized that businesses should consider individual differences in environmental attitudes and behaviors when implementing GHRM practices.

**Keywords:** green human resource management; green psychological climate; environmental green behavior; environmental sensitivity; altruism; hotel employees

## 1. Introduction

In many sectors, the protection of natural resources and the environment is considered of paramount importance for the continuity of businesses. As a consequence of this awareness, the concept of environmental sustainability has become an important issue for policymakers, academicians, and industry implementers. At the same time, researchers agree that the causes of environmental degradation, such as resource scarcity, increased pollution, and loss of biodiversity, have deep roots in human behavior [1]. In this context,

environmental sustainability has become the major concern of decision-makers in the twenty-first century, and as a consequence, new alternatives to traditional human resource management (HRM) have been introduced [2].

Human awareness levels are fundamental to the adoption of more advanced environmental practices [3,4]. In general, environmental sustainability at the organizational level is to a great extent considered to depend on ecological behavior at the individual level [5]. Therefore, to minimize the negative environmental impacts of activities within organizations, there is an evident need to understand and shape the behaviors of employees. Based on these notions, the role of green human resource management (GHRM) in affecting green employee behavior in the workplace has emerged as a subject of study [6]. GHRM is crucial for the effective application of green strategies and environmental management practices [1,7] and may positively contribute to the environmental sustainability of an organization [8].

Human capital is of strategic importance for the success of firms in the tourism and hospitality industry for which activities necessitate interaction and communication with customers and are largely dependent on skills [9]. Creating green employees who understand, appreciate, and implement green initiatives and pursue green goals through GHRM should take an important place in determining the management policies and strategies of tourism businesses [10]. The existence of a green organizational climate implies that the employees who adopt the same environmental values will exhibit green employee behavior at every stage of their jobs. The formation of environmental values and the shaping of the organizational climate through the locomotive effect of GHRM practices has become an important subject to be investigated in the hospitality sector in which the human factor is considered to be the most important input. It is assumed that such an environmental background that will be formed in the business will directly affect the behaviors of the employees. Predicting employee behaviors in the service sector, besides solving many problems, is expected to contribute to the effective and efficient execution of processes.

In this study, the hospitality sector has been preferred to determine the consequences of GHRM practices. The main reasons for this can be explained as follows. GHRM practices have been known to lead to greater efficiency, lower costs, retention of qualified employees, and better employee engagement for businesses [6]. By their nature, hotel businesses are also among the businesses that have high employee turnover rates [11]. In this sense, various strategies have been implemented to reduce employee turnover in businesses and it is observed that human resource management practices have developed and spread over time. Furthermore, environmental factors such as deforestation, loss of biodiversity, ozone depletion, global warming and climate change, and unsustainable use of natural resources have posed a threat to the whole world. Unfortunately, the tourism sector and hotel management within this sector stand out among all sectors for high water consumption, large volumes of indoor waste, and air quality problems [12]. Moreover, the rise of environmental pressures from the market and consumers and the introduction of new regulations and laws have increased the awareness and experience of the tourism sector, including hotels, in addressing environmental issues [13,14]. Therefore, it is of critical importance to retain environmentally-conscious employees in order to fulfill environmentally-conscious functions in the hotel industry and accordingly strategic human resource policies to be structured [15]. In fact, it should be accepted as a key point to create a continuous environmentalist understanding within the business through GHRM practices and to nurture the pro-environmental behaviors of the employees in the workplace. Therefore, the effects of GHRM practices on GPC and also indirectly on EGB have been considered as a problem that is worth examining.

The studies argue that employees need to establish a relationship with themselves to be able to adapt to GHRM practices and/or a green behavior system [16]. In this context, how the environmental sensitivity and altruism levels of employees are integrated into an organizational green behavior system is a point that needs to be examined.

The study has aimed to examine green practices and their results in the human resources department, which is of vital importance for hotel businesses. In this context, the effect of green human resources practices on employees' perceptions of psychological green climate has been examined within the scope of the study. In addition, the effects of green climate perceptions of hotel employees on green behavior have been evaluated. Furthermore, unlike other studies in the literature, the study aims to determine the moderator role of environmental sensitivity and altruism on the effect of green climate perceptions on green behavior. It is thought that the results of the study will provide insights into the relationships between GHRM practices and employees' green behaviors. Taken together, it can be stated that the current study will fill this gap by examining how GHRM leads to positive outcomes at the organizational and individual (i.e., employee) levels.

## 2. Conceptual Model

### 2.1. Green Human Resource Management and Green Psychological Climate

In recent years, organizations have been subjected to environmental pressure coming from consumers, markets, and laws. The need to meet the growing demands of the external environment has led to an increase in environmental awareness in organizations [14]. It is observed in the literature that the studies on green policies in organizations have in–tensified since the 1990s [17–19]. This intensification has drawn attention to the need for the support of human resources to implement environmental practices. In this regard, Renwick et al. [20] developed a GHRM model that underlines the relationship between HRM and a firm's green performance. The previous studies emphasize that human resource practices are critical for the implementation and maintenance of environmental management systems [7,21,22].

Human resource management in the tourism sector, as in all sectors, has used the terms "environmental human resource management", "sustainable human resource management" and "green human resource management" to associate human resource management with environmental management [1]. GHRM, which plays an important role in achieving environmental goals [23], also empowers organizations with environmentally conscious, committed, and competent employees who can help minimize their carbon footprint through efficient and effective use of available resources, including telecommunication tools, reduced paper printing, task sharing, and video conferencing [24].

Masri & Jaaron [25] define GHRM as using human resource management (HRM) practices to strengthen environmentally sustainable practices and increase employee commitment to environmental sustainability issues. Harb & Ahmed [10] assert that GHRM is directly responsible for the creation of green employees that understand, appreciate, and implement green initiatives and consider the term of the green employee as a product of GHRM. In this respect, it can be seen that many studies on GHRM practices have focused on employee behaviors [6,14,26].

The greening process of human resources includes a wide range of policies and practices oriented towards protecting the environment, such as green recruitment and selection, green training, green performance management, green pay and rewards, and green participation [27]. In the literature, it is observed that GHRM practices have been measured in various ways, either multidimensionally [28] or unidimensionally [8,29]. Among these measurement models, the model whose dimensions are called "green recruitment and green selection," "green training," "green performance management," and "green pay and rewards" has been frequently preferred in studies [30–32] and this model has also been used in our study.

It is essential to mention briefly the GHRM practices examined in the aforementioned studies. The first of these is to ensure that GHRM has recruitment strategies that aim to attract employees with similar environmental values and beliefs to the organization in order to be an effective force in revealing the green behaviors of employees.

Environmental training, also called green training, stands out as one of the main ways in which human resources support environmental management and this is another

GHRM practice that has received considerable attention from academicians [29,30]. Green performance and green rewards dimensions include the evaluation of employees on the basis of their environmental performance and compensation and non-monetary incentives to achieve the targeted performance. Development, performance, and reward practices that take into account individual environmental performance; effective training programs that improve environmental awareness, attitudes, skills, and knowledge can be given as examples for these strategies [1,7].

In addition to these traditional human resource practices, the importance of creating a greener organizational culture with the support of HRM [33] has been emphasized through the GHRM. The corporate environmental strategy gives hints about what kinds of behaviors are expected and valued by organizations from their employees, and this in turn leads to a psychological green climate, which is defined as the perceptions of employees [34]. If employees perceive that the conservation of the natural environment is appreciated, rewarded, and motivated in their organization, then a deep individual psychological sense of green climate will arise and be strengthened [35]. As a result, the employees will perform higher levels of voluntary environmental behaviors [36].

In order to determine the relationship of GHRM practices with various variables such as green supply chain management [24,37], organizational citizenship behavior [14], green behavior [38], financial sustainability [10], corporate social responsibility [39]; environmental performance [40], it has been the subject of numerous studies based on different sectors such as automotive [41,42], manufacturing [6,24,25], education [43], and health [2,38]. There are also studies that examine the relationships between the variables that are subject to the study. For example, in the studies conducted in the manufacturing sector, the effect of GHRM on green organizational climate has been determined [6,44]. Despite the significant increase in GHRM-related publications [1,6,24,28,42,45], the number of studies conducted in the hospitality industry is relatively limited. When the studies carried out in the context of tourism have been examined, it can be stated that the publications are mostly accumulated in the last five years [8,14,32,46,47]. Furthermore, there are no studies examining the impact of GHRM practices on GPC in the context of hotel employees. In light of this information, the following hypothesis has been developed.

**H₁.** *GHRM practices have a positive and significant effect on GPC.*

**H₁ₐ.** *Green recruitment and green selection have a positive and significant effect on GPC.*

**H₁ᵦ.** *Green training has a positive and significant effect on GPC.*

**H₁ᵪ.** *Green performance management has a positive and significant effect on GPC.*

**H₁ᵩ.** *Green pay and rewards have a positive and significant effect on GPC.*

*2.2. Psychological Green Climate and Environmental Green Behavior*

The employees may be reluctant to engage in environmental behaviors as most sustainable activities are costly and onerous for individuals [48]. A discrepancy between environmental policies and actual behaviors can be attributed to the lack of behavioral approaches to the individual motivations of employees [49]. It is possible for individuals to search for reasons when performing a behavior. This situation indicates that with the existence of a psychological green climate, individuals will make sense of their behaviors and associate them with reasons [50]. The behavioral HRM literature concedes that HRM may not directly influence employee behavior; instead, its impact is transmitted through various underlying mechanisms [51]. Psychological climate can be considered as one of the mechanisms providing this link.

Psychological climate is one of the main determinants of human behavior [52]. In the literature, it is accepted as a dominant contextual antecedent that affects individual attitudes and behaviors in the field of organizational and environmental psychology [34]. In this sense, psychological climate determines the organizational policies, practices, pro-

cedures, and values observed in the workplace and are considered a result of employees' social interactions [53]. The concept, in short, has been defined as the individual-level perceptions of the work environment [54] or the general perceptions of employees about the organization [55].

In the literature, green climate has been described as the climate for businesses that achieve sustainable goals by implementing a set of pro-environmental policies [56,57]. In light of all these conceptual explanations, GPC may be defined as the perception of an individual regarding an organization's pro-environmental policies, processes, and practices that reflect its green values.

It is recommended that employees should internalize and interpret the HRM practices and policies of the organization and in turn form their perceptions regarding the organization and its values [58]. The climate literature suggests that employee behavior is largely influenced by employee perceptions of the organization [59]. A positive psychological green climate facilitates employees to transform their green behavioral intentions into voluntary environmental behavior [36]. Therefore, environmentally friendly employees will perform more environmental behaviors in the workplace as they observe that the organizational climate sincerely welcomes environmental behaviors [60].

The green behavior of employees (EGB) has been defined as the actions and behaviors that can be improved by engaging employees who are interested in or contribute to environmental sustainability [61]. For this reason, EGB involves activities such as turning off the lights when leaving the office (i.e., saving energy), using teleconferencing facilities instead of going to meetings (i.e., using resources efficiently), organizing documents electronically rather than printing them (i.e., avoiding waste), printing drafts on junk paper (i.e., recycling), and declaring leaks in the bathroom (i.e., saving water) [62].

Sabokro et al. (2021) [44] found that green organizational climate had an impact the on green behaviors of employees in their research in the manufacturing sector. According to the research conducted by Norton et al. (2014) [57], it was concluded that the perceived presence of organizational environmental policies affected employee behaviors. In fact, the study found a relationship between organizational environmental policies and proactive green behavior mediated by green climate. According to Zhou et al. (2018) [63], employees have similar perceptions through a green organizational climate, which helps to achieve the sustainable goals of the firm through improved environmental behaviors and actions. Similarly, studies have shown that GPC strongly influences the in-role and extra-role/voluntary green behaviors of employees [6]. Both GPC and EGB are constructs that exist at the personal level; therefore, GPC is arguably the closest predictor variable of green behaviors [44]. On the basis of these issues, it can be asserted that a psychological green climate has an effect on the green behaviors of employees. The hypothesis that has been developed in this context is as follows.

**H₂.** *Psychological green climate positively and significantly affects the green behaviors of employees.*

*2.3. The Moderator Effect of Environmental Sensitivity and Altruism on the Relationship between Green Psychological Climate and Environmental Green Behavior*

The term altruism refers to a state of motivation whose ultimate goal is to improve the welfare of another person [64]. It is known that the definition of opposite expressions facilitates the understanding of some concepts. Therefore, while explaining the concept of altruism, it will also be useful to explain its opposite, the concept of egoism. Egoism is defined as a motivational state the ultimate goal of which is to increase one's own welfare [64]. Altruism was derived from the Latin word "alter" meaning "other" [65]. This term was first coined in the nineteenth century by the French philosopher and pioneer of Positivism, Auguste Comte, as an antithesis to egoism and was introduced into English by George Henry Lewes in 1853 [66].

Altruistic values are associated with the collective welfare of society and the biosphere rather than trying to serve the individual [67]. This relationship draws attention to the fact that the concept of altruism should also be evaluated from an environmental perspective.

Due to the stable nature of individual values [68], the concept of altruism is becoming a topic of increasing interest in the context of sustainability. It should be considered that altruistic values can predict individuals' environmental beliefs, behaviors, and preferences [69]. In this context, it is considered important to investigate the importance of altruistic values in the relationship between GPC and EGB. Therefore, the present study considered the concept of altruism as a moderator in the proposed model.

In a broad sense, altruism is mostly related to moral and ethical values in which individuals take selfless actions for others [70]. Fehr and Fischbacher (2003) [71] also stated that altruism is critical for guiding behavior and that an altruistic minority can eventually pressure a selfish majority to cooperate. Altruistic behaviors, which naturally arise as a result of experiences of connecting and coming together, are encouraged by values that affect groups and teams rather than individuals [72]. It is thought that these values will be disseminated through the psychological climate to be created in the organization. Thus, it is thought that the possibility of employees exhibiting environmentalist behaviors will increase with the green organizational climate that can be created and the influence of altruistic individuals on other employees. In the literature, there are studies examining the relationship between altruistic behavior and different variables in hotel employees. It is seen that the concept is associated with variables such as status competitiveness, optimism [73], organizational citizenship behavior [65,74], job satisfaction, and turnover intention [66]. Azila-Gbettor (2022) [74] examined the moderating role of the concept of altruistic work value. However, there is no study examining the moderating role of the concept of altruism in the effect of GPC on EGB. Considering this fact, the hypothesis to be tested has been formed as follows.

**H3.** *Altruism has a moderator role in the effect of GPC on EGB.*

Chawla (1998) [75] defines environmental sensitivity (ES) as an interest in learning about the environment, being concerned about the environment, and having a tendency to act toward protecting it. Environmentally conscious individuals have a basic appreciation and interest in the natural environment. In this regard, it was found that there is a close link between ES and the development of pro-environmental behavior [76]. Environmental awareness is considered to be related to the interest in the environment and exhibiting behaviors to protect it, and it is believed to have a very important function in ensuring sustainability [77]. In the literature focusing on the tourism sector, ES is mostly associated with the attitudes of local people [78,79] and tourist behaviors [76,80]. However, it is thought that the environmental sensitivities of employees are of great importance for the development of an environmentalist hotel management approach and the realization of environmentalist practices by hotel businesses. In this respect, the concept of ES has been chosen as a moderator variable between GPC and EGB variables. In the literature, there are studies measuring the differences in ES levels between children and adolescents; women and men [81]. In addition, Cheng & Wu (2015) [76] examined the relationship between ES and environmentally responsible behavior from the tourist perspective. However, no studies have been found to examine the moderator role of ES in the effect of GPC on EGB. The hypothesis that has been formed in this respect is as follows.

**H4.** *ES has a moderator role in the effect of GPC on EGB.*

## 3. Methodology

### 3.1. Research Instrument

The GHRM has been measured with five different scales, which are GPC, EGB, ES, and altruism, through the questionnaire technique. In order to measure GHRM, the scale developed by Jabbour (2011) [28] consisting of 15 items and four sub-dimensions has been used. The 5 statements for GPC were compiled from previous studies in the literature [40,52]. The 5 statements related to EGB have been obtained from Igbal et al. (2018) [82] and Sabokro et al., (2021) [44]. The 4 statements related to ES have been adapted

from Cheng & Wu (2015) [76]. Finally, three statements to measure altruism have been adapted from the study of MacKenzie et al. (1993) [83]. Since the native languages of the scales used in the study were English, the process recommended by Brislin (1976) [84] has been applied. In this context, firstly, the scales were translated into Turkish by a linguist, and secondly, they were translated back into English by a different linguist, thus providing a two-way control. As a result of the translation, it was observed that the two-way translation provided similar outcomes and it was decided that content validity was ensured. Each statement in the scales used was rated using a 5-point Likert scale (1 = strongly disagree − 5 = strongly agree).

### 3.2. Sampling and Data Collection

The tourism sector can cause pressure on natural resources and affect them negatively due to its direct contact and dependence on the environment [85]. For these environmental reasons, the tourism sector was taken into consideration in order to determine the effects of the concepts subject to the research problem on each other, and in this context, employees of hotel enterprises, which is a branch of the tourism industry, were preferred for field research.

The convenience sampling method was preferred since the exact number of employees working in the hotels could not be estimated. There are 215 hotels with tourism-establishment-certificate in Manavgat. In contrast, there is no statistical information about how many people are employed in these hotels as they are seasonal. The internationally accepted method accepts that one staff member can be employed for every two beds [86]. In this context, it has been deemed convenient to take the bed capacity as a basis for determining the sample number. According to the latest data, the bed capacity of the hotels in the Manavgat region was determined as 183,706. [87] the simplified formula of Yamane (1967) [88] has been used for the number of people to be reached. Accordingly, it has been considered necessary to reach at least 384 employees. Within the scope of the study, a pilot study has been conducted to measure the reliability and comprehensibility of the scales before the actual data collection. In the pilot study, the questionnaire form has been applied to 47 hotel employees. The results showed that the Cronbach alpha values of the constructs in the scales were 0.70 and above [89] and the expressions in them were understandable. Based on the results of the pilot study, it has been decided that the actual data can be collected, and a convenience sampling method was preferred in the data collection phase. In this regard, a questionnaire was applied to 442 employees working in five-star hotel businesses.

### 3.3. Common Method Bias

Response-enhancing techniques have been applied because of the potential high risk of common method bias prevalent in research conducted in social sciences [90]. In order to minimize common method bias, each questionnaire was prepared with a cover page. On the cover page, the following statements were included: 'Participation in this research is voluntary', 'There are no right or wrong answers to all statements', and 'Responses to the questionnaires will be used for scientific purposes only'.

According to the data of the Turkish Ministry of Culture and Tourism (2022) [91]; the highest tourist density in the Antalya region is observed in July and August. Therefore, July 2022 has been preferred for data collection. Of the 442 completed questionnaires, 17 have been excluded due to missing data and 425 questionnaires have been subjected to analysis.

### 3.4. Data Analysis

Several procedures have been implemented before the process of determining the inter-relationship effects in the collected data. First, each questionnaire form was numbered and transferred to the SPSS program. Secondly, a three-stage data screening process was applied in the SPSS program. In the first stage, Mahalanobis distance was evaluated to determine the extreme values. As a result of the evaluation, 19 questionnaire forms were found to contain outliers and were excluded from the analysis (Mahalanobis' D (32) > 0.001).

In the second stage, it was examined whether there was a multicollinearity problem. As a result of the examination, it was decided that there was no multicollinearity problem since the values in all constructs were below 5 for the VIF value and above 0.10 for the Tolerance values [92]. In the third stage, the kurtosis and skewness values of the data were examined and it was found that the values were between −1.5 and +1.5. The results obtained indicate that the data show a normal distribution [93].

As a result of the convincing results given above, the AMOS program has been used to test the structural model. In addition, Process macro (Hayes, 2018; model 1) [94] has been preferred in order to determine the moderator effects.

### 3.5. Findings

3.5.1. Demographic Profile

The demographic characteristics of the employees who participated in the survey have been given in Table 1. In this context, 62.1% of the employees are male. When the age ranges of participants are reviewed, 31% of the participants are between the ages of 18 and 25. In addition, 64% of the employees are single. When the educational status of the employees is evaluated, it is determined that the majority of the employees (46.8%) are associate's/bachelor's degree graduates. Finally, it is determined that 53.7% of the employees have 1–5 years of professional experience.

**Table 1.** Demographic profile.

|  |  | **n** | **%** |
|---|---|---|---|
| Gender | Male | 252 | 62.1 |
|  | Female | 154 | 37.9 |
| Age | 18–25 | 126 | 31.0 |
|  | 26–34 | 146 | 26.0 |
|  | 35–45 | 106 | 26.1 |
|  | 46–54 | 22 | 5.4 |
|  | 55 and above | 6 | 1.5 |
| Marital Status | Single | 146 | 36.0 |
|  | Married | 260 | 64.0 |
| Education Level | Primary School | 46 | 11.3 |
|  | Secondary School | 140 | 34.5 |
|  | Associate's/Bachelor's Degree | 190 | 46.8 |
|  | Graduate School | 30 | 7.4 |
| Occupational Experience | 1–5 years | 218 | 53.7 |
|  | 6–10 years | 90 | 22.2 |
|  | 11–15 years | 58 | 14.3 |
|  | 16–20 years | 28 | 6.9 |
|  | 21 and above | 12 | 3.0 |

3.5.2. Confirmatory Factor Analysis Regarding the Structural Model

The confirmatory factor analysis, which is the stage before the path analysis, was applied to test the hypotheses developed based on the research purpose. The results of the confirmatory factor analysis can be seen in Table 2. The first value that should be examined in the analysis is the factor loadings of the items in each construct [95]. As a consequence of the preliminary evaluation, a total of four items, two items in the GPC scale (*All employees are encouraged to save the energy within the workplace; The managers emphasize on reduction of scraps during production*), one item in the EGB scale (*I feel responsible for the environment*) and

one item in the environmental sensitivity scale (*I care about the impact of my living habits on the natural environments*) were excluded from the analysis since their factor loadings found to be below 0.50. Following the second evaluation, it was determined that all factor loadings in the remaining 28 items were 0.50 and above. At the same time, the statements in each construct were significant at the $p \leq 0.001$ level. In contrast, the goodness of fit values obtained are at acceptable levels ($\chi2 = 1130.792$, df = 316, $\chi2/df = 3.578$, NFI = 0.885, IFI = 0.914, TLI = 0.899, RMSEA = 0.080, CFI = 0.914). Based on these results, it has been concluded that the data obtained provide structural model validity.

The Cronbach alpha values in each construct were examined with respect to construct reliability. The reliability values that are presented in Table 2 vary between 0.793 and 0.943. Since these values are above the limit determined by the literature, it can be concluded that each scale meets the reliability requirement [95]. Moreover, the minimum CR value has been determined to be 0.754 and the minimum AVE value has been determined to be 0.573. As a result of the findings, it has been decided that convergent validity and composite reliability are provided [96].

The discriminant validity of each construct in the study has been examined within the context of confirmatory factor analysis and the results of these analyses have been shown in Table 3. When the results of the table are evaluated, it can be determined that the square root of the AVE value of each construct is higher than all the values in the relevant row. These results indicate that the construct provides discriminant validity [97].

### 3.5.3. Hypothesis Tests

The process of determining the path coefficients, which is the second step of the study, was started after obtaining satisfactory results from the confirmatory factor analysis. The goodness of fit values that were determined in the path analysis was found to be within acceptable limits ($\chi2 = 684.773$, df = 191, $\chi2/df = 3.585$, NFI = 0.896, IFI = 0.923, TLI = 0.906, RMSEA = 0.079, CFI = 0.923).

When the results of the hypotheses have been analyzed, the effect of green recruitment and selection, which are the sub-dimensions of GHRM, on GPC is insignificant. Accordingly, hypothesis $H_{1a}$ has been rejected ($p > 0.05$). In contrast, green training ($\beta = 0.16$, t = 2.830, $p < 0.05$), green performance management ($\beta = 0.35$, t = 4.600, $p < 0.001$), and green pay and reward ($\beta = 0.49$, t = 9.298, $p < 0.001$), which are other sub-dimensions of GHRM, have a significant and positive effect on GPC. In light of these results, hypotheses $H_{1b}$, $H_{1c,}$ and $H_{1d}$ have been accepted. Another hypothesis of the study is oriented at determining the effect of green psychological climate on environmentally green behavior. When the path coefficients have been examined, it can be seen that green psychological climate has a significant and positive effect on environmentally green behavior ($\beta = 0.51$, t = 9.336, $p < 0.001$). As a result, $H_2$ has been accepted.

The results of the regression model that was developed to determine the moderating effect can be seen in Table 4. Considering the table values, it is observed that the moderator role of altruism perception in the effect of GPC on EGB is significant ($\beta = 0.06$, 95% CI [0.002, 0.117], $p < 0.05$). Furthermore, while the effect is low for employees with a low perception of altruism ($\beta = 0.11$, 95% CI [0.039, 0.184]), the intensity of the effect is higher for employees with a high perception of altruism ($\beta = 0.21$, 95% CI [0.090, 0.333]). The details of the altruism moderator variable have been displayed in Figure 1.

In a similar respect, the moderator role of environmental sensitivity in the effect of GPC on EGB has been examined and it has been found that ES has a moderator role in the intensity of the effect of GPC on EGB ($\beta = 0.10$, 95% CI [0.006, 0.150], $p < 0.05$). In fact, as the perception of environmental sensitivity decreases, the intensity of the effect of green psychological climate on employee green behavior decreases ($\beta = 0.30$, 95% CI [0.205, 0.405]) and as the perception of environmental sensitivity increases, the intensity of the effect increases ($\beta = 0.44$, 95% CI [0.296, 0.590]). The results regarding the details of the moderator effect have been presented in Figure 2.

**Table 2.** The results of Confirmatory Factor Analysis.

| Factors/Items | Standard Loading | t-Value | $R^2$ | CR | AVE | CA |
|---|---|---|---|---|---|---|
| **Green Human Research Management** | | | | | | |
| *Green recruitment and selection (GRS)* | | | | 0.912 | 0.770 | 0.929 |
| GRS1 | 0.837 | | 0.70 | | | |
| GRS2 | 0.818 | 20.18 * | 0.66 | | | |
| GRS3 | 0.938 | 25.35 * | 0.88 | | | |
| GRS4 | 0.913 | 24.26 * | 0.83 | | | |
| *Green training (GT)* | | | | 0.890 | 0.619 | 0.889 |
| GT1 | 0.801 | | 0.64 | | | |
| GT2 | 0.858 | 19.08 * | 0.73 | | | |
| GT3 | 0.780 | 16.97 * | 0.60 | | | |
| GT4 | 0.724 | 15.44 * | 0.52 | | | |
| GT5 | 0.765 | 16.56 * | 0.58 | | | |
| *Green performance management (GPM)* | | | | 0.918 | 0.791 | 0.919 |
| GPM1 | 0.924 | | 0.85 | | | |
| GPM2 | 0.915 | 29.45 * | 0.83 | | | |
| GPM3 | 0.826 | 23.92 * | 0.68 | | | |
| *Green pay and reward (GPR)* | | | | 0.875 | 0.701 | 0.879 |
| GPR1 | 0.869 | | 0.75 | | | |
| GPR2 | 0.822 | 19.97 * | 0.68 | | | |
| GPR3 | 0.820 | 19.92 * | 0.67 | | | |
| **Green Psychological Climate (GPC)** | | | | 0.819 | 0.603 | 0.793 |
| GPC1 | 0.784 | | 0.61 | | | |
| GPC2 | 0.852 | 16.58 * | 0.72 | | | |
| GPC3 | 0.686 | 13.67 * | 0.47 | | | |
| **Environmentally Green Behavior (EGB)** | | | | 0.754 | 0.573 | 0.859 |
| EGB1 | 0.819 | | 0.67 | | | |
| EGB2 | 0.755 | 17.11 * | 0.57 | | | |
| EGB3 | 0.643 | 13.92 * | 0.41 | | | |
| EGB4 | 0.799 | 18.36 * | 0.63 | | | |
| **Environmental Sensitivity (ES)** | | | | 0.943 | 0.848 | 0.943 |
| ES1 | 0.906 | | 0.82 | | | |
| ES2 | 0.922 | 30.37 * | 0.85 | | | |
| ES3 | 0.935 | 31.82 * | 0.87 | | | |
| **Altruism (ALT)** | | | | 0.911 | 0.774 | 0.909 |
| ALT1 | 0.899 | | 0.81 | | | |
| ALT2 | 0.920 | 28.65 * | 0.84 | | | |
| ALT3 | 0.818 | 22.40 * | 0.67 | | | |

\* $p < 0.001$.

**Table 3.** Discriminant Validity Results.

| Factor | 1 | 2 | 3 | 4 | 5 | 6 | 7 | 8 |
|---|---|---|---|---|---|---|---|---|
| 1. GRS | 0.877 [a] | | | | | | | |
| 2. GT | 0.533 | 0.786 [a] | | | | | | |
| 3. GPM | 0.040 | 0.060 | 0.889 [a] | | | | | |
| 4. GPR | 0.091 | 0.014 | 0.384 | 0.837 [a] | | | | |
| 5. GPC | 0.130 | 0.080 | 0.145 | 0.507 | 0.776 [a] | | | |
| 6. EGB | 0.082 | 0.062 | 0.450 | 0.674 | 0.464 | 0.756 [a] | | |
| 7. ES | 0.107 | 0.105 | 0.315 | 0.454 | 0.453 | 0.505 | 0.920 [a] | |
| 8. ALT | 0.070 | 0.061 | 0.427 | 0.625 | 0.483 | 0.702 | 0.649 | 0.879 [a] |

GRS: Green recruitment and selection, GT: Green training, GPM: Green performance management, GPR: Green pay and reward, GPC: Green Psychological Climate, EGB: Environmentally Green Behavior, ES: Environmental Sensitivity, ALT: Altruism; [a] Square root of the AVE.

**Table 4.** Moderated Effect Result.

| | | | | | **Employee Green Behavior** | | |
|---|---|---|---|---|---|---|---|
| | | | | | $\beta$ | **Confidence Interval** | |
| **H3** | | | | | | Min. | Max. |
| Green psychological climate (X) | | | | | 0.18 ** | 0.113 | 0.289 |
| Altruism (W) | | | | | 0.40 ** | 0.151 | 0.663 |
| X.W (Interaction) | | | | | 0.06 ** | 0.002 | 0.117 |
| $R^2$ | | | | | 0.65 | | |
| Altruism | $\beta$ | S.E. | t | LLCI | ULCI | | |
| Low: | 0.11 ** | 0.03 | 3.03 | 0.039 | 0.184 | | |
| Middle: | 0.15 * | 0.04 | 3.59 | 0.068 | 0.234 | | |
| High: | 0.21 * | 0.06 | 3.42 | 0.090 | 0.333 | | |
| | | | | | Employee green behavior | | |
| | | | | | $\beta$ | Confidence Interval | |
| **H4** | | | | | | Min. | Max. |
| Green psychological climate (X) | | | | | 0.50 * | 0.219 | 0.320 |
| Environmental sensitivity (W) | | | | | 0.21 ** | 0.092 | 0.334 |
| X.W (Interaction) | | | | | 0.10 ** | 0.006 | 0.150 |
| $R^2$ | | | | | 0.33 | | |
| Environmental sensitivity | $\beta$ | S.E. | t | LLCI | ULCI | | |
| Low: | 0.30 * | 0.05 | 5.99 | 0.205 | 0.405 | | |
| Middle: | 0.36 * | 0.05 | 6.80 | 0.259 | 0.469 | | |
| High: | 0.44 * | 0.07 | 5.93 | 0.296 | 0.590 | | |

\* $p < 0.001$; \*\* $p < 0.05$.

In accordance with the results of the model, both $H_3$ and $H_4$ have been supported. Overall, in light of the findings, the coefficients of the whole structure have been presented in Figure 3.

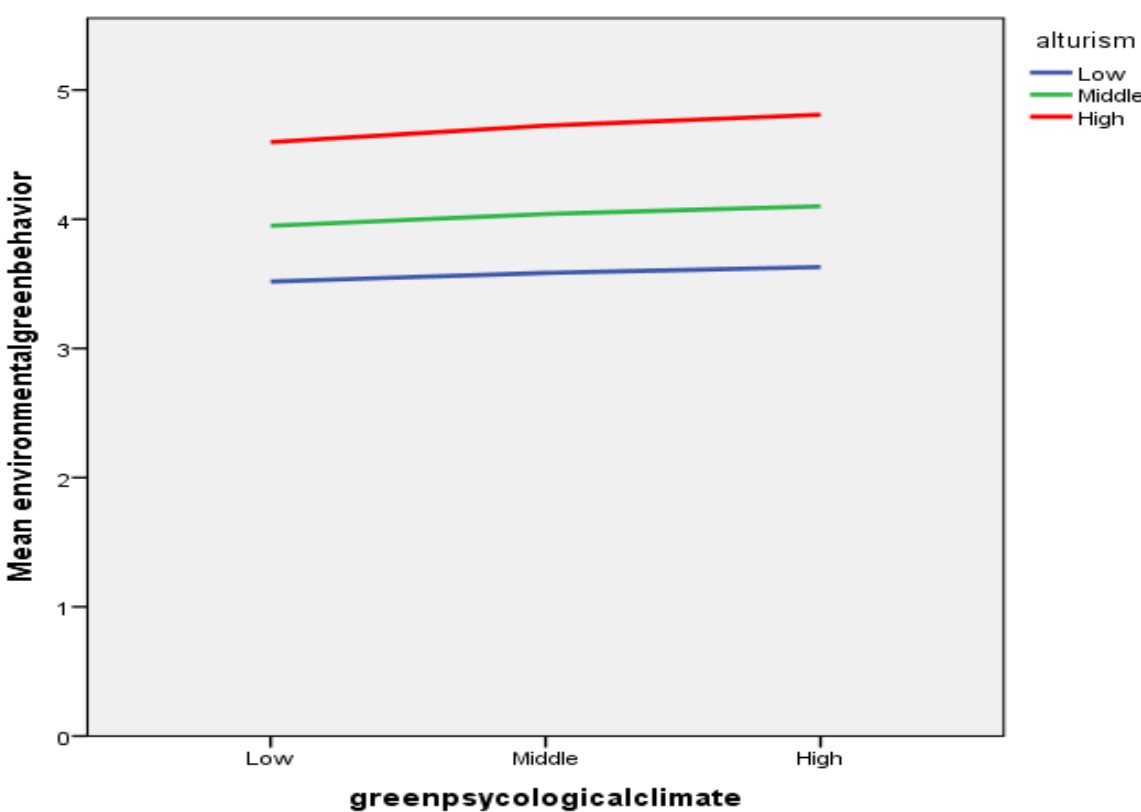

**Figure 1.** ALT as a moderator to GPC-EGB.

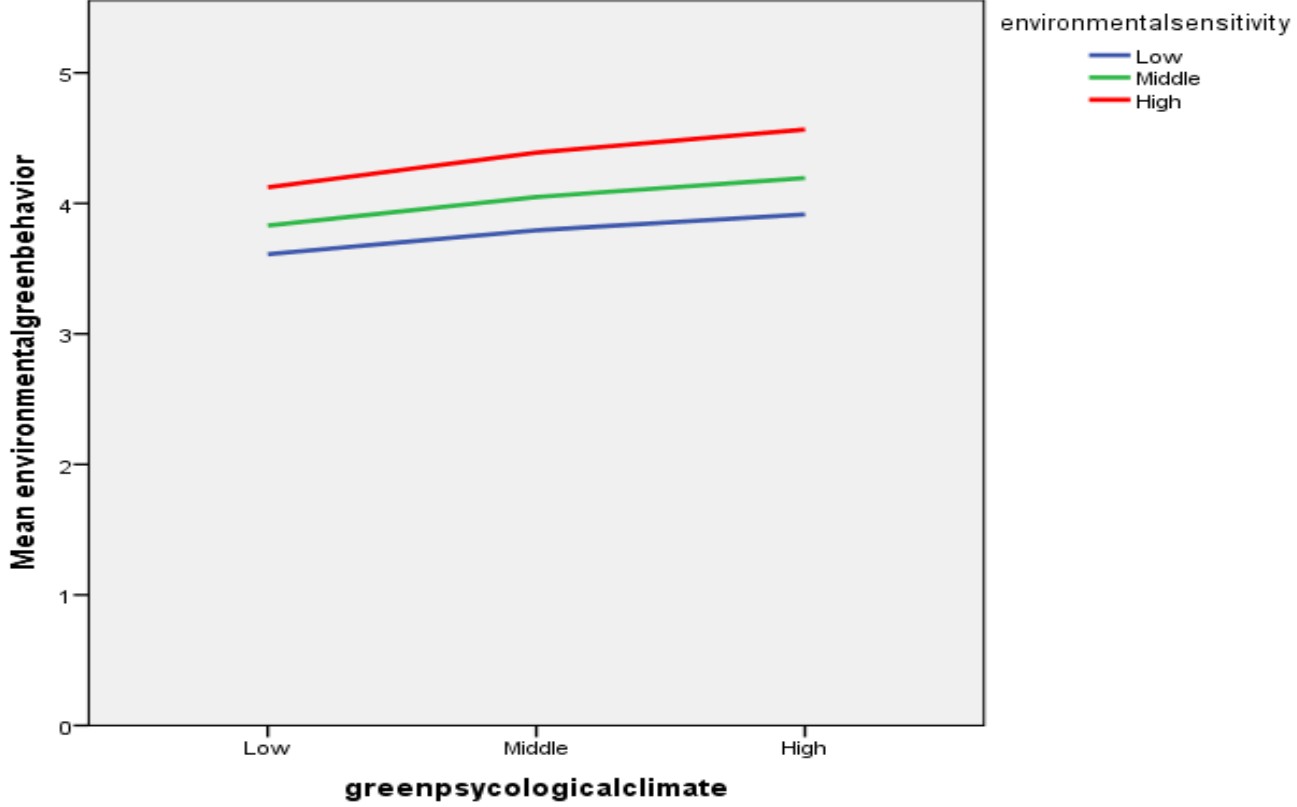

**Figure 2.** ES as a moderator to GPC-EGB.

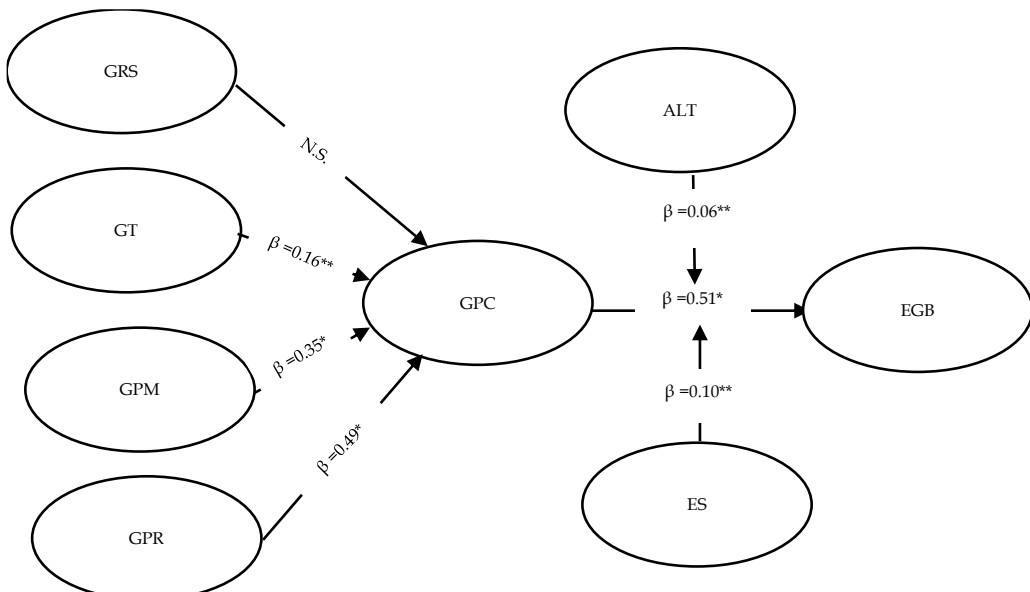

**Figure 3.** Structural Model Coefficients. (GRS: Green recruitment and selection, GT: Green training, GPM: Green performance management, GPR: Green pay and reward, GPC: Green Psychological Climate, EGB: Environmentally Green Behavior, ES: Environmental Sensitivity, ALT: Altruism, * $p = 0.001$, ** $p < 0.05$).).

The findings concerning the hypotheses results have been presented in Table 5.

**Table 5.** Hypothesis Test Results.

| Hypothesis | Results |
|---|---|
| Green recruitment and selection ⟶ Green Psychological Climate | Rejected |
| Green training ⟶ Green Psychological Climate | Supported |
| Green performance management ⟶ Green Psychological Climate | Supported |
| Green pay and reward ⟶ Green Psychological Climate | Supported |
| Green Psychological Climate ⟶ Environmentally Green Behavior | Supported |
| Green Psychological Climate × Environmental Sensitivity ⟶ Environmentally Green Behavior | Supported |
| Green Psychological Climate × Altruism ⟶ Environmentally Green Behavior | Supported |

## 4. Discussion and Implications

GHRM is a relatively new concept that emphasizes the importance of promoting environmental sustainability in the workplace through human resource practices such as training, development, performance management, and employee commitment [2,22,27,29,44,47]. GHRM is also a set of practices that businesses can adopt to promote environmental sustainability, such as implementing energy efficiency measures [98,99], promoting the use of environmentally friendly products [100], and encouraging employees to reduce their carbon footprint [101,102]. By adopting GHRM practices, businesses can encourage their employees to be more environmentally conscious and engage in environmentally friendly behaviors.

The results have indicated that GHRM practices are positively related to GPC, and the degree to which employees perceive their workplace as environmentally friendly. This finding is in line with the research of Sabokro et al. (2021) [44]. Only the H1$_a$ has been rejected. The findings regarding the first hypothesis of the study, the rejection of green recruitment and selection and the acceptance of training and performance management are fully in line with the findings of the research carried out by Nisar et al. (2021) [103] in which

374 employees participated. In this respect, it can be stated that the reason for the negative relationship between the practices of the businesses in the green recruitment and selection process and GPC is that the related businesses are learning organizations and GHRM practices develop environmental understanding, attitude, and GPC after the employees start working. Yet, it has been concluded that the training and proper management of employees regarding environmental issues by GHRM can support the development of GPC within the organization. In addition, GPC is positively related to EGB, which is the degree to which employees engage in environmentally friendly behaviors at work ($H_2$). Saborko et al. (2021) [44] examined the moderator role of GPC in the relationship between GHRM and EGB and determined that GPC had a positive effect. In another study, Tahir, R., Athar, M. R., & Afzal, A. (2020) [104] examined the effect of GPC on EGB and concluded that it had a positive effect. Moreover, in the study conducted by Saeed et al. (2018) [105] in which 347 employees participated, it was found that GPC had a positive moderator effect on the relationship between GHRM and EGB, which is consistent with the findings of this study. These findings are in accordance with the second hypothesis of our study. Therefore, it has been observed that the more developed the GPC attitudes and practices in organizations, the more positive the increase in behaviors related to protecting and sustaining the environment within the framework of EGB.

The unique aspect of the study has revealed that ES and altruism play a positive moderator role in the relationship between GPC and EGB ($H_3$–$H_4$). Although there is no study similar to $H_3$, Cheng and Wu (2015) [76] examined the moderator role of ES in the relationship between environmental knowledge and environmentally responsible behavior and concluded that there is a positive effect. Similarly, Bala, R., Singh, S., & Sharma, K. K. (2023) [106] found that ES positively mediates the relationship between environmental knowledge and environmental behavior intention. Tiwari (2022) [107] found a positive effect between altruism and behavior intention within the context of the theory of planned behavior. In particular, it can be asserted that individuals with high ES and altruism are more likely to engage in environmentally friendly behaviors when they adopt GPC practices. In other words, employees who perceive their workplaces as environmentally friendly are more likely to engage in environmentally sustainable behaviors in the workplace.

This study has important implications for businesses seeking to promote environmental sustainability in the workplace. Businesses can create a culture of environmental sustainability by adopting GHRM practices [108] and encouraging environmentally friendly behaviors among their employees. Moreover, by targeting employees with high levels of ES and altruism, businesses can potentially increase the effectiveness of their GHRM initiatives. However, it is important to note that the data obtained in the study has some limitations, such as the use of the data and a relatively small and specific sample. Further research is needed to confirm these findings and investigate the potential generalizability of the results to other environments and populations.

### 4.1. Theoretical Implications

The theoretical implications of this study transcend the specific context of the research and have broader implications for the fields of sustainability and human resource management. The study has highlighted the importance of environmental management practices in promoting environmental sustainability in businesses. The findings have demonstrated that businesses should prioritize the development and implementation of GHRM practices that support employees to adopt more sustainable behaviors. Secondly, the study has suggested that ES and altruism may increase the effect of GPC on EGB. This finding also suggests that it is important for organizations to take individual differences into account in shaping the effectiveness of GPC and the approaches of employees toward sustainable green management in the organization. Businesses will be able to better tailor their practices to support sustainable behaviors and consequences by recognizing and adapting to individual differences in ES and altruism.

Third, this study has underlined the importance of GPC in directing EGB. A positive GPC can promote a sense of environmental responsibility among employees, motivating them to adopt more sustainable behaviors [44,109]. This finding has significant implications for sustainability-oriented executives who aim to foster change in businesses. By creating a positive GPC, the executives can help promote pro-environmental behaviors among their employees, which in turn can support the achievement of the sustainability goals of the businesses.

In summary, this study has contributed to the ever-growing body of research on sustainability and human resource management. The findings have emphasized the importance of environmental management practices, individual differences, and psychological climate in promoting sustainable behavior in organizations. These insights can inform the development of more effective sustainability strategies in terms of both employee engagement and organizational performance.

### 4.2. Practical Implications

The study also emphasizes the significance of incorporating sustainability considerations into GHRM practices, beyond compliance with environmental regulations. This may include designing job roles and responsibilities by taking sustainability into account [110], providing training on environmental issues [111,112], and aligning performance evaluations [40] and rewards [35] with sustainability goals. This can help create a GPC and encourage employees to engage in environmentally friendly behaviors. Businesses could also promote ES and altruism among their employees by raising awareness about environmental issues, providing opportunities for employees to engage in environmentally friendly activities, and recognizing and rewarding environmentally responsible behavior. This may help to reinforce the positive effects of GHRM practices on GPC and in turn the positive effects of GPC on EGB. Businesses can foster a green corporate culture by developing and communicating a clear environmental mission and values and embedding them in the policies, practices and decision-making processes of the business. This may help to create a strong sense of common purpose and commitment to environmental sustainability among employees.

As a counterpart to positive practices, businesses should identify and handle potential barriers to implementing GHRM practices, such as lack of resources [113], resistance from employees or management [114,115], and competing priorities. This may necessitate making changes to the structures, systems, and culture of businesses and developing effective communication and training programs.

Lastly, the study has also suggested that businesses can contribute to broader sustainability goals by promoting environmentally friendly behaviors among their employees. By reducing their environmental impact and promoting sustainable practices, businesses could help mitigate the negative effects of climate change and other environmental issues [116,117]. This can have benefits not only for the business but also for society and the planet as a whole.

In order to understand the impact of GHRM practices on different types of businesses and different cultural contexts, conducting further research is required. Future research may also examine the role of other moderators such as individual values and beliefs in the relationship between GHRM and environmental behavior.

### 5. Limitations and Future Research Directions

#### 5.1. Limitations

The study, which has examined the relationship between GHRM practices, GPC and EGB, and also examined the moderator effects of ES and altruism, provides valuable information, but it is important to consider some of its limitations when interpreting these findings. The first limitation of the study is its generalizability. The study has been conducted on a specific sample of employees in five hotel establishments in the Manavgat district and this sample may not be representative of other businesses. This means that the

findings may not be valid in other contexts or environments. Another limitation is that the study relies on self-reported data, which may be subject to bias or error. For instance, the participants may have given socially desirable responses or may have had difficulty accurately recalling their behaviors or attitudes.

The study has only focused on a limited set of variables (GHRM, GPC, EGB, ES, altruism), which may not fully reveal the impact of GHRM on environmental outcomes and environmental behaviors. Finally, the study has only examined ES and altruism as potential moderators and has not considered other potential moderators that may influence the relationships between GHRM practices, GPC and EGB. In this respect, future research could improve this study by addressing some of these limitations and further exploring the complex relationships between GHRM practices, GPC and EGB.

*5.2. Future Research Directions*

In this study, although the influence of GHRM on GPC and GPC on EGB has been examined, there may be other contextual factors regarding these relationships. For example, physical environment [118] (e.g., office design, lighting, temperature), social environment [119] (e.g., relationships with coworkers, supervisor support), and broader cultural and social contexts [120] (e.g., government policies, media coverage) may influence the effect of GHRM on GPC and GPC's on EGB. Future research could also investigate how these contextual factors interact with GHRM practices to shape the attitudes and behaviors of employees toward the environment. This research has primarily focused on the consequences of GHRM practices at the employee level. Nevertheless, it is important to consider the perspectives of other stakeholders such as customers, suppliers, and investors. Future research may focus on how GHRM practices influence the perceptions and behaviors of these stakeholders and how their involvement in environmental issues influences employees' GPC and environmental behaviors.

The study has specifically focused on the relationship between GHRM practices and GPC and EGB. However, there may be other GHRM practices that may influence environmental consequences. Therefore, future research could compare the effectiveness of GHRM practices in promoting environmental sustainability with other HRM practices such as diversity management [121], employee empowerment [122], and corporate social responsibility [44].

Technology and innovation have the potential to play an important role in promoting environmental sustainability. Thus, future research could explore how technology and innovation intersect with GHRM practices to promote GPC and EGB. For instance, how can GHRM practices be adapted to support the development and adoption of green technologies, and how can technological innovations be used to improve GHRM practices?

In addition, the study has been conducted in a particular cultural context. Future research might analyze whether the findings are valid in other cultural contexts and whether the relationships between GHRM, GPC, and EGB differ across cultures.

Our current research has focused on the moderator roles of ES and altruism. Future research may also examine other individual and contextual variables such as personal values, social norms, and perceived organizational support that could potentially moderate the relationships between GHRM, GPC, and EGB. Finally, while identifying the relationships between GHRM, GPC, and EGB, the study has not examined other constructs and variables that explain these relationships. Future studies could focus on dependent/independent variables such as green employee engagement, green job satisfaction, and green organizational citizenship behaviors.

The moderator roles of altruism and ES concepts have been determined through this study. However, in future studies, examining whether these variables have mediating effects in the relevant model will provide a different perspective on the literature.

All in all, there seem to be many different directions that future research can take in exploring the relationship between GHRM, GPC, and EGB. The researchers may continue

to examine these issues and help businesses develop more effective strategies to promote environmental sustainability and handle the challenges of climate change.

## 6. Conclusions

The study has revealed that there is a positive effect of GHRM practices on GPC and GPC has positively affected EGB. This implies that the employees who perceive their businesses as implementing GHRM practices are more likely to have a positive perception concerning GPC and positive GPC is more likely to engage in pro-environmental behaviors. These findings indicate that businesses can use GHRM as a tool to promote pro-environmental behavior among their employees and contribute to environmental sustainability.

In the research, it has been also found that ES and altruism have a positive moderator effect on the relationship between GPC and EGB. This indicates that more environmentally conscious and altruistic individuals are more likely to benefit from GHRM practices, leading to a more positive GPC and greater participation in the EGB. For this reason, businesses should consider individual differences in ES and altruism when implementing GHRM practices to maximize their effectiveness. In this way, they can effectively promote environmental sustainability and have a positive effect on the environment.

**Author Contributions:** The preparation of this research paper has been decided and done by all of the authors in cooperation, yet each author has specifically contributed to the paper. The project administration and language supervision have been done by H.K., O.Y. has carried out the analysis and focused on the hypotheses' testing process and the scales of the paper. A.T., G.S.E. and F.U. have given their effort in the preparation of the original draft by conducting an extensive literature review and they also formed the research hypotheses. A.A. and A.K. have contributed to completing the conclusion, discussion, implications, and limitations parts. Furthermore, during the preparation process, the authors used an internal auditing system and supervised each other to avoid any possible drawbacks. All authors have read and agreed to the published version of the manuscript.

**Funding:** This research received no external funding.

**Institutional Review Board Statement:** This study was conducted according to the guidelines of the Declaration of Helsinki and was approved by the following ethics committees: Akdeniz University—Social and Human Sciences Scientific Research and Publication Ethical Committee (Ref: 06/108).

**Informed Consent Statement:** Informed consent was obtained from all participants involved in the research.

**Data Availability Statement:** The data analyzed during this study are available on request from the corresponding author.

**Conflicts of Interest:** The authors declare no conflict of interest.

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
