# Peer review of "Effect of Green Human Resource Management on Green Psychological Climate and Environmental Green Behavior of Hotel Employees: The Moderator Roles of Environmental Sensitivity and Altruism"

_sustainability, doi:10.3390/su15076017_

Round 1
Reviewer 1 Report
the study is overall good and have meaningful for the practice of modern HRM. the evidence and method is also effective and good to support the hypothesis of the paper. the presents of the results is clear and well structure. please if possible have some more discuss about the findings of the study and its different from other previous studies.
Author Response
Dear esteemed reviewers;
As a result of your detailed evaluation of our manuscript, we have highlighted the necessary changes, corrections, and additions to the manuscript in line with the suggestions you made to us. Moreover; we have prepared a detailed report on all these changes made to the manuscript and uploaded it to the system to be examined by you. Thank you very much for your contribution to our study.

Reviewer 2 Report
· Line 21: This article reports the findings of…
· Line 27: AMOS software, not program.
· Line 51: humanitarian aspects are not apparent in this sentence.
· Line 76: the studies- aspected to cite few sources than one.
· Line 80-92 is vague, and I recommend the authors be precise on ‘what to research’.
· Lines 101, 113, 115: Check the citation and content accuracy.
· Line 105-108: not straightforward, specifically Academician association. Please rewrite.
· Lines 279-282: Forward-backward translation done, but how was the content validity at this stage concluded? Any referral from the content experts than translators.
· Line 302-303: Face-to-face questionnaire; this term is vague.
Introduction
· This paper’s introduction does not directly focus on the green practices, antecedents, and outcomes to the hotel industry and relevant employees. It is too broad at this point. The research is interesting despite not being unique. Perhaps have some significance to the body of knowledge. The specific factual problems/issues within the hotel industry that need such research are not in the introduction. Research objectives should be in a measurable approach.
Literature review
· The authors narrated the literature with little critique. These can be further improved to build the strength of the paper. The moderating effects of the selected variable in the context of hotel employees, green practices, and inclusion will be advantages for this paper. Why environmental sensitivity and altruism are premiers of this research and not others? Is it possible for these two variables to play a mediating role? It will enrich the writing.
Method
· The instrument used in the study is well described. The population of the survey is missing. How was the sample determined not clear? Data collection procedures are not clear. Statement of ethics encouraged for inclusion in this paper.
Findings
· Include a discussion on Common Method Bias (CMB). The analysis could have been easy and simplified if the authors had used SmartPLS software and run PLS-SEM. However, using the current statistics to test his hypotheses is good. Tables reporting the findings must be well-organised, using a standard table format. Please look into this matter. The Strenght of the measurement model is acceptable, and the structural model is too.
Discussion
· I would suggest that this part can accommodate a few previous empirical findings; nevertheless, the authors’ expression is more vital in discussing the implications of the hypotheses testing. The final piece of this paper should be reduced and not sound like a lengthy thesis writing. I encourage the authors to be precise in their contributions (theory and practice). Limitations and future research are acceptable, but again don’t write in a thesis context – precision is a must.
· I encourage the authors to check the writing styles carefully, in any case paraphrasing software used. A second round of professional proofreading could help to reconfirm the quality of the writing.
· I have randomly checked the list of references against the in-text referencing, which is somewhat accurate. This field of study is still young and might have been surveyed and reported elsewhere. Thus, the latest literature incorporated in the text will ensure the paper remains current and supports the findings.
Author Response
As a result of your detailed evaluation of our manuscript, we have highlighted the necessary changes, corrections, and additions to the manuscript in line with the suggestions you made to us. Moreover; we have prepared a detailed report on all these changes made to the manuscript and uploaded it to the system to be examined by you. Thank you very much for your contribution to our study.

Reviewer 3 Report
The paper is well structured and deals with a very current topic. Also, recent and relevant sources of scientific literature were used. However, the theoretical part of the work is much better written than the analysis of the obtained research results. Thus, there are no significant remarks, except that the paper's title is too long and too complicated, and it is not clear from the proposed title of the paper that it is research conducted in the hotel industry. This title does not indicate which area of the economy it refers to. Therefore, before publishing the paper, the title should be simplified, and shortened, and the title indicates that the title that the paper refers to the hotel industry. Also, the abstract should be changed so that it is immediately clear that the research is related to the hotel industry.
Author Response
Dear esteemed reviewer(s),
As a result of your detailed evaluation of our manuscript, we have highlighted the necessary changes, corrections, and additions to the manuscript in line with the suggestions you made to us. Moreover; we have prepared a detailed report on all these changes made to the manuscript and uploaded it to the system to be examined by you. Thank you very much for your contribution to our study.

Round 2
Reviewer 2 Report
The authors have made significant changes as required, and the paper looks better than before. Please proofread the paper again. The final decision depends on the editors.